# Multicomponent Home-Based Training Program for Chronic Kidney Disease Patients during Movement Restriction

**DOI:** 10.3390/ijerph18073416

**Published:** 2021-03-25

**Authors:** Olga López-Torres, Celia Azpeitia-Martínez, Marcela González-Gross, Dace Reihmane, Amelia Guadalupe-Grau

**Affiliations:** 1ImFINE Research Group, Health and Human Performance Department, Facultad de Ciencias de la Actividad Física y del Deporte-INEF, Universidad Politécnica de Madrid, 28040 Madrid, Spain; c.azpeitiam@alumnos.upm.es (C.A.-M.); marcela.gonzalez.gross@upm.es (M.G.-G.); amelia.guadalupe@upm.es (A.G.-G.); 2Department of Human Physiology and Biochemistry, Riga Stradiņš University, Dzirciema Street 16, LV-1007 Riga, Latvia; dace.reihmane@rsu.lv

**Keywords:** renal diseases, exercise, lockdown, hemodialysis, social distancing

## Abstract

Both intradialytic and out-of-clinic exercise programs (EP) have been proven to be a safe and effective way to increase fitness levels in end-stage chronic kidney disease (CKD) patients. The actual COVID-19 pandemic situation has forced the suspension of EP offered in hemodialysis centers in many countries; as well as all activities considered as “non-essential” (i.e., sport facilities and fitness centers). Therefore, there is a high risk that movement restrictions would promote physical inactivity and its associated diseases in CKD patients; especially those undergoing domiciliary confinement situations. Given the importance for CKD patients’ overall health to maintain exercise levels and reach physical activity recommendations, the aim of this Protocol was to design a personalized, well-structured, multicomponent physical EP that CKD patients can safely follow at home. We also aimed to provide an initial fitness evaluation tool that allows patients to adapt the EP to their fitness level. Current general exercise recommendations for people living with chronic conditions have been analyzed to develop the present home-based EP proposal.

## 1. Introduction

End-stage chronic kidney disease (CKD) is a serious public health problem characterized by the presence of kidney damage or an estimated glomerular filtration rate (eGFR) less than 60 mL/min/1.73 mt^2^, persisting for three months or more [1]. This state of progressive loss of kidney function affects medical, social, and economic systems globally [2]. Hypertension and diabetes are the most common causes of CKD [3], whereas age and gender are additional contributors [4,5]. 

Functional capacity of renal failure patients is 50–70% lower compared to their healthy counterparts, which implies a reduction in flexibility, muscle strength and resistance, coordination, and cardiovascular endurance [6], lower quality of life [7], and increased depression and anxiety [8]. Low functional capacity is one of the most important factors of mortality in older adults in general and especially in CDK patients [9]. It is well known that aging also reduces physical fitness [10]. Therefore, older adults with CKD would be a population at high risk. Sedentary behavior increases with age [11] and hemodialysis sessions extend long sitting periods in CKD patients, resulting in reduced physical activity (PA) (arbitrary units) level in this population by 3.4% per month [12]. 

Exercise has beneficial effects on the cardiovascular and musculoskeletal system, as well as on the mental state and quality of life, and no negative adverse effects on the CKD patient’s disease severity have been reported [4]. Regular exercise is beneficial for patients in all CKD stages, with the majority of evidence being for end stage kidney disease (ESKD) [13]. Therefore, controlled exercise programs (EP) have been introduced in many hemodialysis centers during treatment with positive results [5,10,14]. The guidelines of American College of Sports Medicine (ACSM) for CKD patients recommend 20–60 minutes/day for 3–5 days/week of aerobic exercise training at mild to moderate strength (40–60% of VO2 peak), as well as resistance exercise training at 70–75% of one-repetition maximum for a minimum one set of 10–15 repetitions on 2–3 days/week [15]. Hence, CKD patients must perform physical exercise, both intradialytic and outside sessions. There is no consensus in the scientific literature on type of intervention which results in the greatest overall benefits [16]. Therefore, clinicians do not have clear guidelines on best exercise to prescribe for their patients. Nevertheless, any kind of PA may have benefits on CKD patients’ health and quality of life [7]. In 2013, a position statement on exercise and chronic kidney disease was published by The Australian Association for Exercise and Sports Science (AAESS) [17]. The statement highlights the need of regular exercise for at least 30 min, three times/week which can improve physical fitness, cardiovascular dimensions, and health related quality of life. Furthermore, recommendations described in Clinical Practice Guideline for the Evaluation and Management of Chronic Kidney Disease [18] for CKD patients include aerobic, resistance and flexibility activities for at least 30 min, 5 times per week. Looking at PA guidelines for the general population and older adults specifically [19], they are similar to those for CKD patients in terms of frequency, intensity, and types of PA, which must combine endurance, resistance, flexibility, and balance exercises at least five days/week.

A common perception is that intradialytic exercise is more feasible to implement (although less than 10% of hemodialysis clinics worldwide have EP, and enrollment and adherence in these are generally low) [6] while out-of-clinic programs provide more exercise variety [9] but lower patient safety, compliance, or adherence [16]. Commonly, both options (intradialytic or out-of-clinic programs) use either endurance exercises, such as pedaling [7], or resistance exercise training protocols, such as ankle weights and resistance bands or a combination of both [7]. Some interventions with out-of-clinic programs or peri dialytic exercise (performed in the waiting room) have used weight machines and other materials [20,21], such as elastic bands, dumbbells, or soft balls. Theoretically, out-of-center EP should not have any limitations on the types of activities that can be performed (the patient is not attached to a machine) and concerns about practicing exercise during hemodialysis [16,22] have not been deeply studied or confirmed. Nonetheless, most studies are designed to develop training programs during dialysis in hemodialysis centers while studies using out-of-clinic EP are scarce and focus on three categories: (1) home-based walking programs; (2) resistance training in fitness centers; and (3) combined programs that include aerobic and/or resistance exercise [9,22]. Proposing home-based walking as the only way to improve physical capacity in CKD patients might not reach the expected intensity and/or recommendations [23], and multicomponent training might be needed to achieve an optimal fitness and functionality. In this population, walking should be recommended as unprescribed daily PA outside the clinic and patients should be engaged to reach the recommendations, while performing another fitness program that allows them to improve and maintain physical condition [15]. Programs carried out in the fitness centers might be a good option due to the extensive equipment and constant supervision. However, for the patients it requires travelling to the fitness centers, adapt to a concrete schedule and less flexibility in the training hours. 

Although most of the scientific studies implementing EP in CKD patients were performed during hemodialysis sessions, less than 10% of hemodialysis clinics worldwide have these EP, and enrollment and adherence are generally low [6]. There are several reasons that may hinder the implementation of EP as standard care in hemodialysis centers, for example, lack of knowledge in exercise-planning or expertise to prescribe it by medical staff [24], difficulties to implement exercise due to other work-related obligations [25], or belief that most patients do not want to exercise [26] although patients perceive exercise as positive [10]. Moreover, patients who had witnessed or participated in EP were more positive towards exercising than those who had not [26]. Due to previously mentioned reasons, PA levels in CKD patients remain below recommendations resulting in low physical function in this population [27]. An interdisciplinary approach is needed including prescription of exercise by nephrologists and dialysis professionals, individualizations of EP by exercise professionals, and encouragement and assistance for patients during intra-dialysis exercise provided by staff members performing dialysis. It has been shown that patient’s inactivity status was strongly associated with dialysis staff’s negative attitudes towards patient’s PA [28]. If exercise practice and reaching the PA recommendation could help this population to maintain or improve health, the effort must be done. The combination of home-based and intra-dialytic endurance and resistance training, designed by exercise professionals with the clinician supervision and dialysis staff support, following an adequate progression and safe exercises seems to be the most suitable option providing an opportunity for a greater volume and intensity. 

Endurance training is the most common exercise used in intra-dialytic clinics. Seated pedaling seems to be easy to implement during dialysis, while there are some concerns about resistance exercise especially with the access arm. Recently, it was shown [29] that both aerobic exercise and the combination of aerobic and resistance exercise significantly improved physical function and quality of life of hemodialysis patients, but combined programs had a better effect on physical function. Studies [27] implementing intra-dialytic resistance-training programs usually use ankle weights for leg training and free weights (dumbbells) for upper body training. Non-access arm can be trained during the treatment, while the access arm must be trained before hemodialysis in the waiting room or at home. For the upper body clinicians normally recommend lower intensity resistance exercise and limit lifting heavy loads to avoid problems in the arm with vascular access, despite limited evidence for this risk [16]. Although it is recognized that general health and quality of life of CKD patients would improve with increasing PA and exercise levels, there is little consensus on ways how to reach it the best. The volume and intensity of exercise performed by CKD patients does not reach the recommendations for older adults and chronic disease patients [16,30]. Some studies [23] reported either endurance or resistance training exercises, performed at low-moderate intensities, for a total of 60–135 min of exercise/week, what is far below recommendations in most PA guidelines. Contrary to what is thought, frailty should not be considered as contraindication to exercise, because most of the CKD patients are not too frail to perform resistance exercise of adequate intensity and achieve increase in muscle size and strength [31].

On 11 March 2020, the World Health Organization (WHO) declared a global pandemic caused by a new virus named SARS-CoV2. Many governments adopted home confinement and social distancing to content the virus. Therefore, EP carried out in hemodialysis centers have been suspended. Due to the proven benefits of exercise and PA in CKD patients [32], it is important to maintain exercise levels even during lockdown or social distancing. Moreover, mix-training programs (during hemodialysis in clinical centers and a home-based program for days without hemodialysis) would be optimal for CKD patients. Although the sanitary emergency in Europe improved during the summer, new waves of COVID-19 cases are happening worldwide. More detailed knowledge on the structure of virus, its action mechanism, new treatments, and the development of new vaccines bring hope. However, some preventive actions must be implemented due to the critical health status of CKD patients and the uncertainty of what will happen in the future. Since older adults and people with chronic diseases are at risk population and CKD patients represent both groups, extra care must be taken. As mentioned before, only 10% of hemodialysis centers have any kind of fitness program. If CKD patients cannot perform the EP in hemodialysis centers due to restrictions imposed to deal with the public health emergencies, possibilities to train at home should be provided. Moreover, a global health initiative managed by the ACSM, known as “Exercise is Medicine^®^” [33], aims to make PA and exercise a standard in clinical care. The initiative encourages all health care providers to include EP when designing treatment plans based on the principle that exercise promotes optimal health and is integral in the prevention and treatment of many medical conditions [33]. Those programs should be based on scientific evidence and design by qualified exercise professionals and clinicians. Therefore, the aim of this Protocol is to provide nephrologists, clinical staff, specialized trainers, and CKD patients, especially those over 65 years old, with a home-based physical EP, designed by exercise professionals, following the institutional recommendations for this population, and taking into consideration safety (physical and medical) precautions. This protocol proposal can be used as a guide by clinical staff members for safe implementation of exercise in the treatment program of their patients. To adapt the program to individual physical levels, characteristics, and health conditions, a tool to obtain an initial fitness status is provided. These kind of programs have been published before for other diseases, like diabetes [33]. The home-based training program is designed to be performed in combination with intra-dialytic EP at clinics, when possible, or alone at home, if that is the only possibility due to the sanitary emergency or if the specific center does not have one.

This EP is not an intervention for a research study but proposal of protocol, that allows incorporation of exercise as a part of CKD patient’s lives supervised by health care professionals. Therefore, neither results nor statistical analysis are provided. 

To determine the effectiveness of EP proposal intervention study in future could be performed. The aim of such intervention study would be to test the hypothesis that EP implementation in CKD patients for 6 consecutive months would result in improved functionality, physical fitness and health-related outcomes, as well as in increase in muscle strength and muscle mass.

## 2. Method: Training Proposal

The training program is a simple, basic material (i.e., elastic bands, a chair, or lightweights), structured multicomponent training, designed for hemodialysis patients and performed during confinement (see Appendix A). Concerns about resistance training preformed out-of-clinics include restricted access to proper equipment. Training program described in this paper uses simple and easy to access materials for exercise. The exercise protocol has been developed following the ACSM guidelines [34], the AAESS position statement [17] and Clinical Practice Guideline for the Evaluation and Management of Chronic Kidney Disease [18] and is designed to maintain muscle mass and cardiovascular function, as well as other important capacities such as balance and mobility [15]. All patients involved in this training program must hold medical authorization allowing exercise interventions. Additionally, patients must receive instructions from a qualified specialist (e.g., aexercise professional) to perform the exercises with the correct technique and safely. Before starting the training program, patients should undergo a thorough medical examination [17]. Evaluation of general health condition, nutritional status and fitness level are mandatory prior to taking part in EP, in both, intra-dialytic or out-of-clinic programs [35]. Exclusion criteria are electrolyte abnormalities like hypo/hyperkaliemia, symptomatic achy-arrhythmias or brady-arrhythmias, excess inter-dialytic weight gain (>4 kg since last dialysis or exercise session), unstable dialysis treatment and changing (titrating) medication regime, pulmonary congestion, and peripheral edema [17]. Blood pressure and heart rate must be controlled with a specific device (digital blood pressure monitor, heart rate monitor, bracelet, or similar). If digital devices are not available, heart rate can be measured manually lightly pressing the index and middle fingers of one hand on the opposite wrist just below the base of the thumb or on the side of the neck just below the jawbone, counting the number of beats in 15 s, and multiplying by four. Heart rate must be measured before, during and after training and very strenuous activity must be avoided. Hydration should be adequate to their condition [36]. Due to fluid retention, the home-based fitness program should be performed days after hemodialysis. Patients receiving pharmacological treatment for hypertension should not exercise within two hours after taking the drugs, as it can alter the heart rate. Room temperature must be between 18 °C and 25 °C [4]. Specific technical and safety considerations that must be taken into account during practicing the EP are explained in the Appendix A that includes the training program. Nevertheless, it is crucial that the patients do not feel any pain or discomfort in the fistula or catheter. Any exercise producing discomfort in the fistula or catheter must be avoided or modified.

The protocol consists of 6 warm-up and activation exercises, followed by 10 min of sitting-pedaling or any other similar activity as the endurance training. The main part is a strength circuit of 10 exercises for the lower and upper limbs, followed by another set of aerobic training (10 min of sitting-pedaling or similar) and a final cool-down: 5 min walking (low intensity) with arm mobility. If the person does not have access to a pedaling machine, it can be substituted by walking at medium-high speed or using any other machine such as a static bicycle or a treadmill. The training session can be performed 3–4 times/week, depending on the hemodialysis sessions. Therefore, two different circuits for each fitness level are proposed for the main part, that can be performed alternatively. As an example, if the patient had 3 day/week hemodialysis, he/she should perform the home-based training the other 4 days, but if hemodialysis were 4 days/week, the patient should perform the program 3 days. When training 4 days/week, circuit 1 must be done on day 1 and 3, while circuit 2 on day 2 and 4 of home-based training. When training 3 days, the first week circuit 1 must be performed on day 1 and 3 of training and circuit 2 on day 2 of training. The next week, circuit 2 will be done on day 1 and 3, and circuit 1 on day 2, and so on. Two sets of 8–15 repetitions must be performed from each circuit exercise, with 30–45 s rest between sets, depending on the physical capacity (see Figure 1). The session should be completed with a moderate level of perceived fatigue (5–6 out of 10) (see Appendix A). Elevated heart rate, pain, and discomfort should be avoided [14]. Similar training programs [37,38] have been implemented with no side effects. If the physical fitness and physiological variables are stable, the patient could extend the initial 10 min of endurance training. 

When patient receives medical approval confirming health condition that is suitable for exercise, the patient must perform the physical test battery to determine the initial fitness capacity. The battery tests should be performed by clinic staff when possible, or by themselves at home. According to the final score, the patients will be classified into one of the 3 levels: low, medium, or high fitness. At the same time, different health-related indexes will be used to adjust the final fitness level according to cardiovascular risk factors [39]. The physical fitness battery consists of three validated tests from the senior fitness test [40]: (i) 6 min walking test (6MWT) to evaluate aerobic capacity, (ii) up and go test (UGT), to evaluate agility and dynamic balance and (iii) the sit-to-stand test (STST), to evaluate lower limb strength [40,41]. More details on how to perform these tests could be found in Appendix A. To obtain the final score, the number of low (1 point), average (2 points), and high (3 points) results of the abovementioned 3 tests according to the values in Table 1 (cut-off points for the physical tests) must be counted [40]. Patients will be classified in the first level if score ≤ 3 points, second level= 4–7 points, and third level ≥ 7 points. As example, a 68 years-old man that obtains the following marks for UGT, 6MWT, and STST respectively; 5.4 s (2 points), 750 m (3 points), and 10 repetitions (1 point), would obtain a final score of 6 points, which classify him in the second level (see Table 1).

After obtaining a score in the fitness tests, patients should adjust their level according to the health-related ratios used to evaluate central obesity: (i) body mass index (BMI), (ii) waist circumference (WC), (iii) Waist-to-Hip Ratio (WHR), and (iv) Waist-to-height Ratio (WHtR) (32)]. More details on how to measure these ratios could be found in Appendix A. If patients obtained one or more ratio of very high risk or three or more ratios of high risk, fitness level will be reduced by one, to avoid possible complications or cardiovascular events, as CKD patients have high prevalence of these pathologies. In the previously mentioned example, if the 68 year-old patient would have obtained one of the mentioned values in the very high risk (or 3 in the high risk), his fitness level would have been reduced from second to first level. When handgrip strength (HGST) can be measured in the medical center and the patients know the value, this test will be also used. Cut-off points [41] for the health-related ratios are presented in Table 2.

In order for clinic staff to be able to reproduce the protocol, a Template for Intervention Description and Replication (TIDieR)checklist was created (see Appendix A).

## 3. Conclusions

In conclusion, PA levels in CKD patients are low and exercise should be prescribed as a part of treatment in this population, in order to improve the overall health status and physical condition of these patients. Multidisciplinary approach should be used to promote the exercise as a part of holistic and modern treatment in CKD patients in future, involving medical doctors (e.g., nephrologists, and cardiologists) and health care professionals (e.g., physiotherapists, exercise professionals, renal dieticians, and nurses). Meanwhile, the home-based exercise protocol proposed in this paper follows international PA recommendations and is in compliance with previously published intervention studies on intra-dialytic aerobic and resistance exercise; thus, could be used as substitute for exercise training programs performed in hemodialysis clinics during movement restrictions caused by, e.g., COVID-19.

## Figures and Tables

**Figure 1 ijerph-18-03416-f001:**
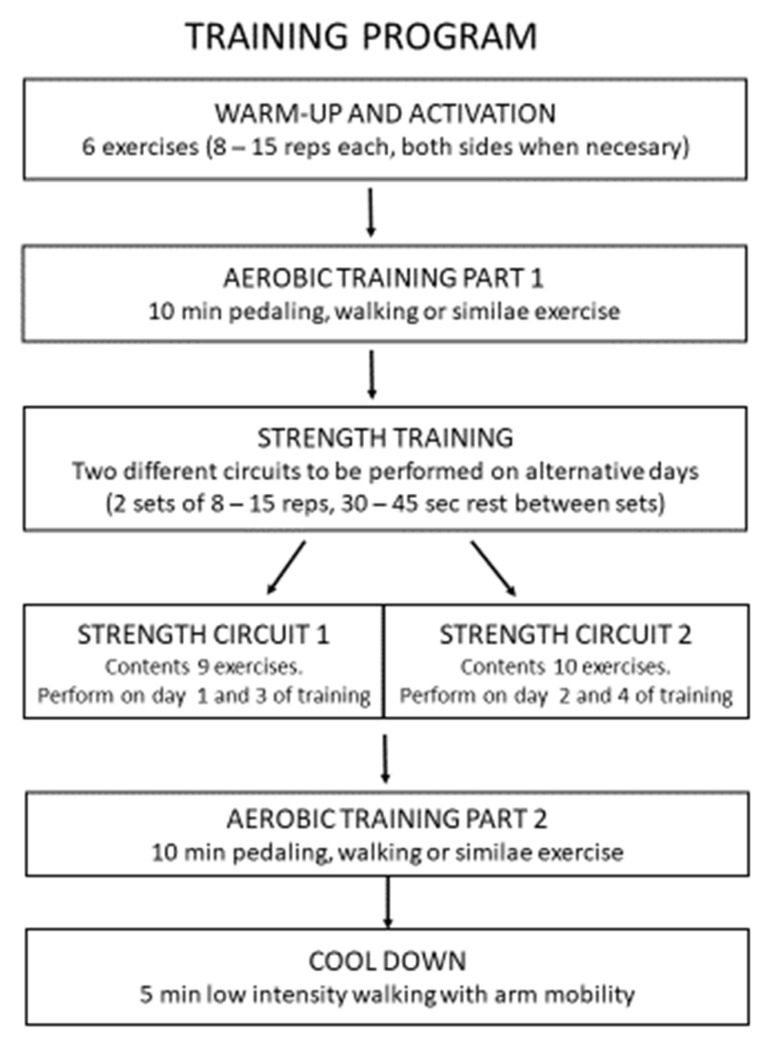
Exercise program description. Reps: repetitions, Min: minutes, Sec: seconds.

**Table 1 ijerph-18-03416-t001:** Classification by fitness tests.

Physical Tests	Sex	Age Group 65–69			Age Group 70–74			Age Group 75–79			Age Group 80–84			Age Group 85–89			Age Group 90–94		
		Low	Average	High	Low	Average	High	Low	Average	High	Low	Average	High	Low	Average	High	Low	Average	High
UGT (sec)	M	≥6.0	5.9–4.3	≤4.2	≥6.3	6.2–4.4	≤4.3	≥7.3	7.2–4.6	≤4.5	≥7.7	7.6–5.2	≤5.1	≥9	8.9–5.5	≤5.4	≥10.1	10–6.2	≤6.1
	W	≥6.5	6.4–4.8	≤4.7	≥7.2	7.1–4.9	≤4.8	≥7.5	7.4–5.2	≤5.1	≥8.8	8.7–5.7	≤5.6	≥9.7	9.6–6.2	≤6.1	≥11.6	11.5–7.3	≤7.2
STST (rep)	M	≤11	12–18	≥19	≤11	12–17	≥18	≤10	11–17	≥18	≤14	10–15	≥16	≤7	8–14	≥15	≤6	7–12	≥13
	W	≤10	11–16	≥17	≤9	10–15	≥16	≤9	10–15	≥16	≤13	9–14	≥15	≤7	8–13	≥14	≤3	4–11	≥10
6MWT (m)	M	≤559	560–700	≥701	≤544	545–680	≥681	≤469	470–640	≥641	≤444	445–605	≥606	≤379	380–570	≥571	≤304	305–500	≥501
	W	≤499	500–635	≥636	≤479	480–615	≥616	≤434	435–585	≥586	≤384	385–540	≥541	≤339	340–510	≥511	≤274	275–440	≥441

Count the number of low (1 point), average (2 points), and high (3 points) results of the 3 tests. Patients will be classified in the first level if score ≤ 3 points, second level= 4–7 points, and third level ≥7 points. UGT: Up and go test, STST: 30 sec sit to stand test, 6MWT: 6 min walking test. M: men, W: women. Sec: seconds, rep; repetitions, m; minutes.

**Table 2 ijerph-18-03416-t002:** Classification by health markers.

Health Markers	Sex	Age Group	Risk Level			
			Low	Medium	High	Very High
BMI			≤18.49	18.50–24.99	25–34.99	≥35.00
WHR	Men		≤0.90	0.91–0.93	0.94–1.03	≥1.04
	Women		≤0.76	0.77–0.84	0.85–0.9	≥1.00
WC (cm)	Men		≤94		94.1–102	≥102.1
	Women		≤80		80.1–88	≥88.1
WHtR	Men		≤0.49	0.50–0.56	0.57–0.63	≥0.63
	Women		≤0.50	0.51–0.57	0.58–0.65	≥0.65
HGST (kg)	Men	AG 60–69	≥40.9	37.1–40.8	26.6–37	≤26.5
		AG > 70	≥35.8	32.2–35.7	22.9–32.1	≤22.8
	Women	AG 60–69	≥24.7	21.8–24.6	16.7–21.7	≤16.6
		AG > 70	≥20.1	16.9–20	10–16.8	≤9.9

BMI: body mass index, WC: waist circumference, WHR: Waist-to-Hip Ratio, WHtR: Waist-to-height Ratio, HGST: handgrip strength test. cm: centimeters, kg: Kilograms, AG: Age group. If patients obtain one or more indices of very high risk or three or more indices of high risk, fitness level will be reduced one level from the initial score.

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
