# Peer review of "Multicomponent Home-Based Training Program for Chronic Kidney Disease Patients during Movement Restriction"

_ijerph, 2021, doi:10.3390/ijerph18073416_

Round 1
Reviewer 1 Report
General comments: Thank you for the opportunity to review this manuscript. Physical capacity in patients with end-stage renal disease is an important public health concern. This work addresses the added challenge of exercise adherence for patents under conditions imposed by the global pandemic.
Please check the wording throughout the manuscript, some editing would improve clarity of the key messages.
Line 53/54: “clinicians do not have clear guidelines on (the) best exercise to prescribe...” Please see this resource:
Smart, NA, Williams, AD, Levinger, I et al. (2013) Exercise and Sports Science Australia (ESSA) position statement on exercise and chronic kidney disease. J Sci Med Sport: 16 (5) pp: 406-11 DOI: 10.1016/j.jsams.2013.01.005
Figure 1: check spelling: necessary, reps, contains vs contents
Table 1: adjust spacing for clarity and include units where appropriate (eg presumably the score in the 6MWT is distance in meters?)
Table 2: ag vs age? And please include units where appropriate (eg grip strength)
It is unclear how the training program would differ for patients with different fitness capacity (as determined by their score on the three physical tests). This calls into question the purpose of the physical tests and the need to adjust the score with the health markers.
The manuscript feels unfinished. It ends without any concluding statements or clear direction as to how the program would be implemented.
Reviewer 2 Report
Review
Multicomponent home-based training program for chronic kidney disease patients during movement restriction
General remarks to the authors
The importance of multicomponent training in aging and many illnesses is clear and fundamental to public health. The main purpose of this Protocol was to design a personalized; well-structured; multicomponent physical exercise program that CKD patients can safely follow at home for during movement restriction due to pandemic. They also aimed to provide an initial fitness evaluation tool that allows patients to adapt the exercise intensity to their fitness level. Current general exercise recommendations for people living with chronic conditions have been analyzed and implemented to develop the present home-based exercise protocol proposal. While I understand the importance of the study, I have some major concerns. First, I suggest including objective and hypothesis of the future study. Also, describe objectively the primary and secondary outcomes, by the way, the PA levels seems to be interesting measure. I suggest using TIDieR checklist: Please report both the intervention and control (I think you need incorporating a control group) using the TIDieR checklist and upload as Supplementary material. It will help readers implement your interventions. Please find an example here: https://static-content.springer.com/esm/art%3A10.1186%2Fs13063-018-2509-7/MediaObjects/13063_2018_2509_MOESM3_ESM.pdf
Statistical procedures, ethical considerations, flow chart, inclusion and exclusion criteria, and type of study are missing.
Minor comments
23 – and
42 – informing that PA has a.u. is meaningless here. Consider removing it.
54-55 - Nevertheless, any 54 kind of PA may have benefits on CKD patients ́ health and quality of life [7].
Consider changing PA for EP, because the previous phrase is leading with exercise, not physical activity.
72-74 – please, consider using a reference to support the statement that home-based walking as the only method of training might not reach the expected intensity.
86-88 – again, the confusion on PA and exercise occurs here. I’d suggest indicating that application of exercise methods (e.g. multicomponent) can improve PA levels improving further the general health of the patients.
128 – 130 – Consider informing in details when the patients should controlling the HR.
Reviewer 3 Report
Please, clarify in more detail the objectives, the method (participants, instruments, procedure, analysis) and the results or conclusions of the study in the summary of the work.
The introduction is timely, adequate, and complete
The section "Method: training proposal" generates many doubts in me. Does not make the proposal clear. The procedure is not clear. I ask and I ask the authors that if it is a proposal, if it is a research project that they do it with greater clarity, with sections, better explained each one of them. Better clarify what you want would be the participants.
It generates many doubts. I don't see an investigation. Rather I see a research project. And if it is a project, it must be formulated much better. The types of analysis should be incorporated. How those expected results would be presented. Even possible discussions of the results
Round 2
Reviewer 2 Report
Dear authors, congrats, at least to me, the paper is ready for publication.
Best wishes, reviewer.
Reviewer 3 Report
the indicated changes have been made